# The Whitening Effect and Histological Safety of Nonthermal Atmospheric Plasma Inducing Tooth Bleaching

**DOI:** 10.3390/ijerph18094714

**Published:** 2021-04-28

**Authors:** Seoul-Hee Nam, Byul Bo Ra Choi, Gyoo-Cheon Kim

**Affiliations:** 1Department of Dental Hygiene, College of Health Science, Kangwon National University, Samcheok 25949, Korea; nshee@kangwon.ac.kr; 2Feagle Co., Ltd., Yangsan 50614, Korea; cbbrstar@naver.com; 3Department of Oral Anatomy, School of Dentistry, Pusan National University, Yangsan 50612, Korea

**Keywords:** nonthermal atmospheric pressure plasma, safety, tooth bleaching, oral soft tissues, histological damage

## Abstract

Various light sources have been applied to enhance the bleaching effect. This study was to identify the histological evaluation in oral soft tissues, as well as tooth color change after tooth bleaching by nonthermal atmospheric pressure plasma (NAPP). Nine New Zealand adult female rabbits were randomly divided into three groups (*n* = 3): group 1 received no treatment; group 2 was treated with NAPP and 15% carbamide peroxide (CP), which contains 5.4% H_2_O_2_, and group 3 was treated with 15% CP without NAPP. Color change (Δ*E*) was measured using the Shade Eye NCC colorimeter. Animals were euthanized one day later to analyze the histological responses occurring in oral soft tissues, including pulp, gingiva, tongue, buccal mucosa, and hard and soft palates. Changes in all samples were analyzed by hematoxylin and eosin staining and Masson’s trichrome. Teeth treated with plasma showed higher ΔE than that obtained with bleaching agents alone. Overall, the histological characteristics observed no appreciable changes. The combinational treatment of plasma had not indicated inflammatory responses as well as thermal damages. NAPP did not cause histological damage in oral soft tissues during tooth bleaching. We suggest that NAPP could be a novel alternative energy source to conventional light sources for tooth bleaching.

## 1. Introduction

Following the initial introduction of tooth bleaching using nonthermal atmospheric pressure plasma (NAPP) [1], many articles have reported that NAPP-based bleaching is much more effective than the conventional methods [2,3,4,5,6,7,8]. Besides the bleaching efficiency, preserving healthy oral tissues without damage is very important during tooth bleaching using NAPP. Nam et al. [9] demonstrated that tooth bleaching using NAPP with 15% HP or CP did not influence the microhardness and the mineral content of dental hard tissues. In addition, the application of NAPP did not result in any structural–morphological and topographic changes in the enamel [10]. Although hard tissue analysis has been reported after tooth bleaching using NAPP, safety analysis on oral soft tissues has not been made so far.

Vital tooth bleaching procedures have been considered one of the most conservative treatments in esthetic dentistry. The most popular bleaching agents are hydrogen peroxide (HP; H_2_O_2_) and carbamide peroxide (CP; CH_6_N_2_O_3_), which diffuse through the organic matrix of enamel and dentin [11]. Tooth bleaching results from the redox reaction between the peroxide-based bleaching agents and the darkened substrate [12]. Currently, tooth bleaching with CP is very popular. Higher concentrations of CP contain higher amounts of HP, and, for this reason, bleaching results are achieved more quickly than when lower concentrations of CP are used [13]. However, tooth bleaching has been reported to cause several adverse effects [14,15,16]. The most common adverse effects are tooth hypersensitivity, gingival irritation, and pulpal damage [17,18]. The use of higher concentrations of bleaching agents can pose a higher risk of tooth and oral soft tissue damage and cause chemical burns when in contact with teeth, gingival tissue, oral mucosa, skin, or eyes [19,20,21].

Various light sources have been applied to enhance the bleaching effect [16]. They accelerate the decomposition of HP into ·OH radical, which makes bleaching a faster reaction [20]. This theory supports that the production of ·OH radical is hastened by temperature increase [22]. OH radical induces oxidative damage on cells and tissues, so that chronical generation of it could lead to specific diseases [23]. Numerous studies have proved that light-activated bleaching did not produce any perceivable color changes [24]. Moreover, the heat generated by light-activated tooth bleaching is capable of diffusing through enamel and dentin and reaching the pulpal space, potentially leading to irreversible thermal damage of the pulp tissue [25]. Thus, the safety of light-activated bleaching is still a matter of controversy, and more evidence is required to make a precise assertion of its effectiveness and safety. Many efforts have been made to increase patient benefits regarding better bleaching efficacy and safe application with recent bleaching systems [25]. Therefore, in order to be a better and safe tooth bleaching method, an effective and nonthermal energy source is desired.

Our previous study has shown that NAPP increased the production of ∙OH radicals and yielded a much better bleaching effect than that of conventional light sources [1,3,5,7], while the tooth temperature was maintained near the body temperature [7]. Furthermore, tooth bleaching with NAPP did not affect the microhardness and mineral content of the tooth [7]. The histological investigation of oral tissues after NAPP treatment is essential for its safe utilization. The 15% CP is used for home bleaching and contains a relatively low HP concentration of 5.4%. Llena et al. [26] reported that CP had a less cytotoxic effect on the human dental pulp stem cells than other products. In our previous paper, the bleaching effect of 15% CP itself was insignificant, but the combination treatment of plasma with 15% CP showed similar results to the bleaching effect of 30% HP [1,4]. The procedure of home bleaching using 15% CP takes a long time to achieve a bleaching effect, but the pain complaints of the operator are relatively small. Therefore, NAPP treatment could reduce the time required for the bleaching effect of 15% CP without the injury to the teeth. However, the histological safety of NAPP has not been fully demonstrated in oral soft tissues such as gingiva, tongue, buccal mucosa, and hard and soft palates. If there is an energy source that effectively induces the tooth color change without causing tissue damages, it would be a useful device for tooth bleaching in clinical practice. The aim of this study was to evaluate the histological changes in the oral soft tissues treated with NAPP and 15% CP to prove that NAPP is safe as well as effective.

## 2. Materials and Methods

### 2.1. NAPP Device

A schematic of the NAPP device was developed (Figure 1). The plasma generating module of the inner stainless and outer copper electrode was made of a ceramic body as a dielectric. The outer electrode is grounded, and a sinusoidal high voltage is applied to the inner electrodes by a high voltage source, which can increase the voltage up to 10 kV with a frequency of 15 kHz. Over 2.8 kHz can generate the plasma in an alumina tube when argon was sued as buffer gas at a low rate of 2 standard liters per minute. The temperature of the NAPP flow at 1 cm from the electrode end was maintained around 35 °C for 10 min [1,3,4,5,6,7]. The distance of treating subjects from the electrodes was kept at 1 cm.

### 2.2. Tooth Bleaching Procedure

Nine New Zealand adult female rabbits, older than six months and ranging in weight from 2.5 to 3.5 kg, were used in this study. Animals were housed individually in separate rabbit cages at the ambient temperature of 20 °C and a 12/12-h light/dark cycle. Animals had free access to drinking water and standard laboratory pellets. At the time of the study, there were no clinical signs of disease in any animal. The protocol was reviewed and approved by the Pusan National University Institutional Animal Care and Use Committee (PNU-IACUC), Pusan, Korea, in 2012 (approval 2012-0086). The overall experimental design of this study is depicted in Figure 2. All rabbits were subjected to general anesthesia, performed with an intramuscular 5:1 mixture of ketamine HCl (50 mg/kg, Ketalasr^®^, Yuhan, Korea) and xylazine (10 mg/kg, Rumpun^®^, Bayer, Korea), and operated on under sterile conditions prior to the bleaching procedure. The nine animals were then randomly divided into three groups of three animals each, according to the treatment. Group 1, the control group, was not treated. Group 2 was treated with NAPP in combination with 15% CP (Kool white 15%, Pac-dent International, Walnut, CA, USA). A uniform 1 mm-thick layer of 15% CP was applied over the labial surface of the upper incisor teeth. During NAPP exposure, the NAPP tip was positioned approximately 1 cm away from the enamel surface. Group 3 was treated with 15% CP without NAPP and left undisturbed. After the 30-min bleaching treatment, the gel was gently wiped with sterile gauze.

### 2.3. Color Changes

The outcome of color change (Δ*E*) was determined by the colorimeter (Shade Eye NCC, Shofu Inc., Kyoto, Japan). The color values were based on the lab color system, CIE (Commission Internationale de L’Eclairage), a widely used tooth color evaluation [27]. The value differences of *L**, *a**, and *b** in each group were measured using Adobe Photoshop CS2 (Adobe Systems, San Jose, CA, USA). The overall Δ*E* for the total difference in the values of *L**, *a**, and *b** were measured and calculated according to the following formula:(1)$$\Delta E=\sqrt{{\left(\Delta L*\right)}^{2}+{\left(\Delta a*\right)}^{2}+{\left(\Delta b*\right)}^{2}}$$

### 2.4. Histological Evaluation

Tissue biopsies were performed one day after bleaching, as it is the optimal time to evaluate any reaction. Rabbits were euthanized, and teeth, gingiva, tongue, buccal mucosa, and hard and soft palates were harvested for histological analysis. All biopsied tissues were placed in different labeled vials containing 4% paraformaldehyde fixative solution at room temperature for at least 48 h. Teeth were decalcified in neutral 10% ethylenediaminetetraacetic acid (EDTA) for four weeks, and the EDTA solution was changed every day to accelerate demineralization. The fixed tissues were embedded in paraffin under vacuum. Serial sections of 4 μm thickness were cut using a rotary microtome, mounted on glass slides, and subjected to hematoxylin and eosin (H&E) and Masson’s trichrome (MST) staining. H&E staining was performed to evaluate epidermal or dermal changes and morphological changes of tissue; MST, the three-color staining in histology, was used to detect collagen fibers. The stained tissues were examined microscopically, in a blinded manner, using a light microscope (CKX41, Olympus, Tokyo, Japan) adapted with a digital camera (Pixel link PL-B686 CU, Canada), at a magnification of ×100, ×200, and ×400. Images were loaded into a computer and processed with Image-Pro Plus 5.1 software (Media cybernetics Inc., Washington, DC, USA).

## 3. Results

### 3.1. Change in Tooth Color

The mean Δ*E* values using colorimeter were shown in Figure 3. The mean Δ*E*± standard deviations (S.D) were 12.36 ± 0.61 and 7.61 ± 0.32 after 30 min treatment with plasma and without plasma, respectively. After 24 h, the mean values of Δ*E* ± SD were 5.60 ± 0.17, and 4.23 ± 0.08 after 1 day of treatment with plasma and without plasma, respectively. An improvement of tooth color was observed at the application of NAPP.

### 3.2. Response of Pulp Tissue

After H&E and MST staining, samples were examined under a light microscope (Figure 4). H&E staining revealed that the three groups of pulp tissues had normal characteristics in all respects, presenting normal cellular structure and configuration. Pulp tissues from groups 2 and 3 were indistinguishable from those of the control, group 1 (Figure 4a). None of the groups showed cellular displacement of odontoblast nuclei into dentin tubules. The odontoblasts were arranged beneath the dentin in the form of one layer with normal size and shape. In addition, none of the pulp tissues presented cellular hyperactivity, collagen disorganization, or intense vascularization, which is evidence of thermal damage and inflammation in MST (Figure 4b).

### 3.3. Effects on Oral Soft Tissues

Tissues of gingiva, tongue, buccal mucosa, and hard and soft palates were stained by H&E after tooth bleaching using NAPP (Figure 5a, Figure 6a, Figure 7a, Figure 8a and Figure 9a). The cornified layer was not separated from the underlying basal layer, and the configuration of both epithelial and connective tissue cell layers was intact in H&E staining. A columnar shape of basal cells with well-defined cell membrane, followed by the cuboidal and squamous shape of the stratum spinosum is observed in all three groups. Additionally, the structures of epithelial and dermal layers in the gingiva, tongue, buccal mucosa, and in hard and soft palates in groups 2 and 3 were consistent with those of the control group, by examination after MST (Figure 5b, Figure 6b, Figure 7b, Figure 8b and Figure 9b). Densely packed collagen fibers were not different from the normal state, and fibroblasts, which were numerous and clear in all samples, were observed scattered throughout the connective tissue. Blood vessels were lined with endothelial cells, and red blood corpuscles were observed in the vascular lumen. All oral soft tissues in each experimental group exhibited histologically normal characteristics without disorganization of collagen fibers, changes of cell morphology, thermal damage, or inflammatory response.

## 4. Discussion

In tooth bleaching treatment, preserving healthy pulp and oral soft tissues are the most important issues. A novel tooth bleaching technique using NAPP has been shown superior in efficacy, but its safety has not been verified. Therefore, if the combination treatment of NAPP and 15% CP is shown to cause no histological damage, NAPP is a safe tool for tooth bleaching.

In clinical tooth bleaching, a high concentration of HP or CP has been recommended to be used with a high-intensity light source. However, the possible adverse effects resulting from tooth bleaching have prompted scientific deliberations in dentistry [28]. The high peroxide penetration into the pulp may result in significant damage to tissue health [29]. Within 5 to 15 min, peroxide penetrates the pulp, where it irritates nerves, consequently producing reversible pulpitis [30], which dramatically inhibits pulpal enzyme activity [31] and causes cytotoxic effects to odontoblast-like cells [32]. A number of reports have demonstrated the deleterious effects of bleaching agents on oral soft tissues [33]. Attia-Zouair et al. [34] revealed that the high concentration of HP in bleaching agents might adversely affect not only the gingival epithelium but also the subepithelial tissue. A form of tissue ulceration or blister may be noted in the gingival tissues at the cervical area, or on the lips or other mucosa, as a distinct white change in the tissue with associated underlying erythema [35]. Consequently, severe dentinal pain is often felt by patients after bleaching [16]. Therefore, the use of bleaching agents with low concentrations of HP is preferred in order to minimize the side effects caused by peroxide penetration. The conventional light sources may affect cell membrane proteins and trigger an autocatalytic reaction, which may result in mutagenesis, carcinogenesis, irreversible cell membrane damage, reduced cell proliferation, and cell death [32]. The high temperature of light-activated bleaching constitutes a serious threat to the vitality of the pulp and oral soft tissues.

The healthy pulp of vital teeth exhibits dentinal fluid flow produced by intrapulpal pressure, cytoplasmatic prolongations of odontoblasts, and other intratubular components [36], which may prevent the diffusion of chemicals through the dentinal tubules [37]. Heat from a light source causes expansion of the liquid in the dentinal tubules and affects the vessels of the pulp [38]. Bowles and Ugwuneri [39] have also shown that the increase of the temperature in the pulp may cause inactivation of several pulp enzymes that are important to the maintenance of physiological cellular functions. The histological studies associated with heat have reported the disappearance of the odontoblastic layer under the treated area and a dense inflammatory infiltrate in the pulps of dogs [40].

Zach and Cohen [41] have shown that pulpal temperature increases of 5.5 °C, 11.1 °C, and 16.6 °C caused 15%, 60%, and 100% irreversible pulp damage, respectively, in *Macaca rhesus*. If the temperature transmitted towards the cervical region leads to an increase exceeding 10 °C, it may cause periodontal injury [42]. Thus, light sources reached a limit temperature for pulp (5.5 °C) and periodontal tissue (10 °C). To preserve the healthy tissues, low thermal conductivity is important to avoid possible thermal damage [43]. Therefore, evaluation of the safety of light-activated bleaching has received considerable attention. In our previous studies, the temperature of the tooth treated with NAPP stabilized near 37 °C, similar to human body temperature, and this state was continuously maintained throughout the tooth bleaching process [1,3,5.7]. The NAPP would be employed with a low and safe temperature.

Additionally, the immune system genes of rabbits are apparently more similar to those of the human immune system than rodent genes [44]. For these reasons, rabbits were used as the animal model. In this study, H&E and MST staining were conducted to determine whether NAPP is safe to pulp tissue and oral soft tissues of rabbits after tooth bleaching using NAPP and 15% CP. In all pulp tissues after tooth bleaching with NAPP, the odontoblasts and endothelial cells were intact, with columnar shape and normal size. No inflammatory response or thermal damage was observed in pulp tissues. Besides pulpal tissues, all analyzed oral soft tissues showed the characteristics of normal tissues with a multi-layer epithelium consisting of keratinocytes, well-organized collagen fibers, and active fibroblasts. No significant difference was found in any of the epithelial and dermal layers after tooth bleaching with NAPP. These results demonstrate that combined tooth bleaching of NAPP and 15% CP does not damage oral tissues, as NAPP does not generate high temperature and quickly dissolves toxic HP to nontoxic ·OH. Because tissue safety in this study was investigated in a short period, it is necessary to investigate it over a long period in the future.

## 5. Conclusions

NAPP has a greater capability for effective tooth bleaching than a low concentration of HP alone. This study provides evidence for the histological safety of the combined action of NAPP and low concentration of CP (15%) in tooth bleaching. This treatment does not induce thermal damage and inflammatory response in pulp and oral soft tissues. The treatment with NAPP will bring further developments in tooth bleaching procedures. NAPP is as it maximizes the bleaching effect and minimizes the potential risks of tissue damage.

## Figures and Tables

**Figure 1 ijerph-18-04714-f001:**
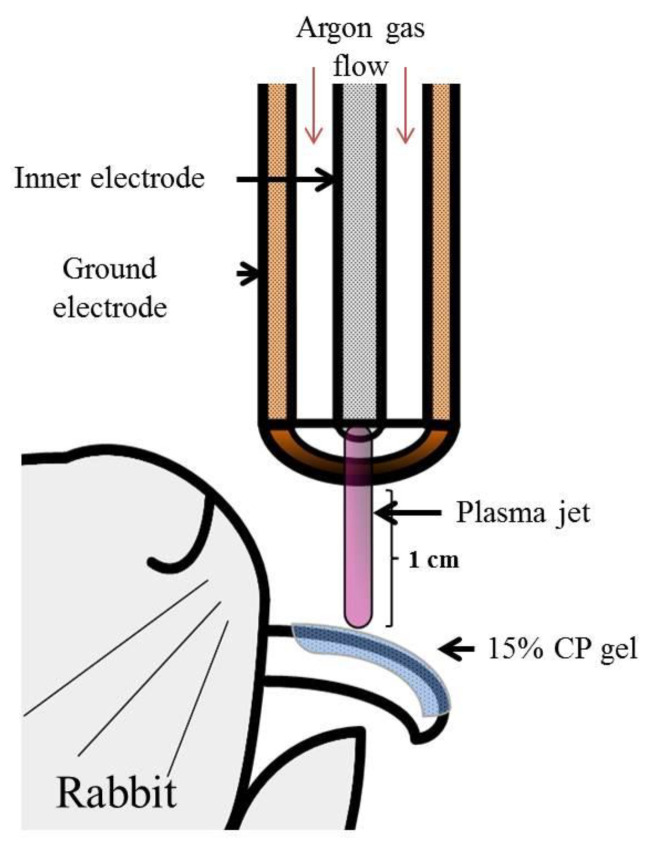
Illustration of the methodology used in the study. Schematic drawing of the NAPP device.

**Figure 2 ijerph-18-04714-f002:**
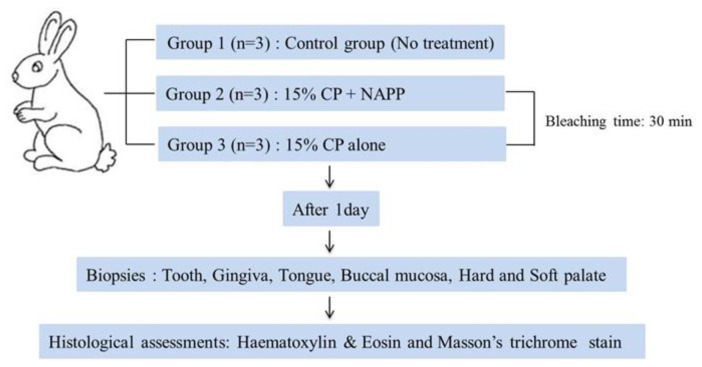
Flow chart of the experimental design for the in vivo study.

**Figure 3 ijerph-18-04714-f003:**
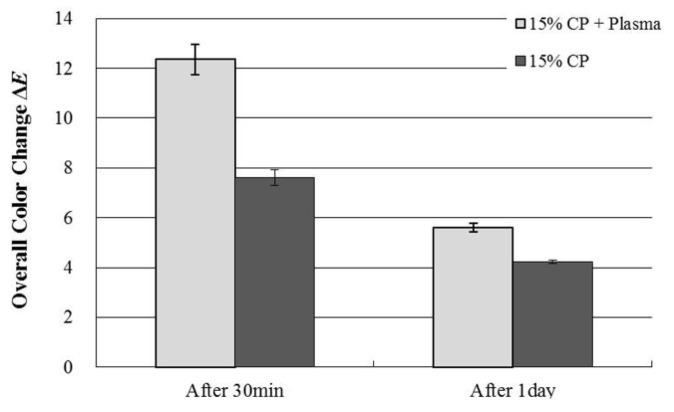
Change in Δ*E* values after tooth bleaching using plasma using Shade Eye NCC colorimeter.

**Figure 4 ijerph-18-04714-f004:**
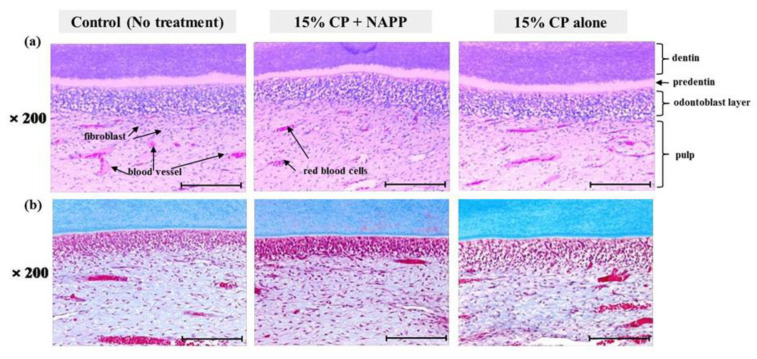
Histological response of pulp tissue to bleaching treatment in each group; (**a**) Representative H&E image and (**b**) MST image. Magnification ×200.

**Figure 5 ijerph-18-04714-f005:**
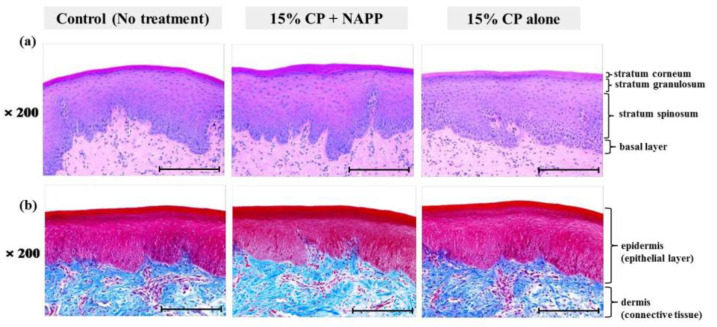
Representative histological image of gingival tissue of each group with (**a**) H&E and (**b**) MST staining. Magnification ×200.

**Figure 6 ijerph-18-04714-f006:**
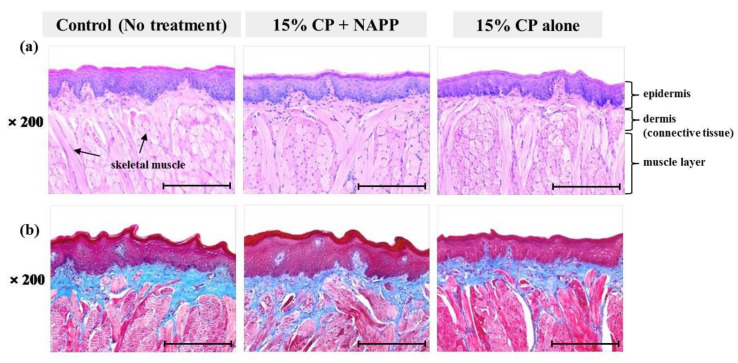
Representative histological image of tongue tissue of each group with (**a**) H&E and (**b**) MST staining. Magnification ×200.

**Figure 7 ijerph-18-04714-f007:**
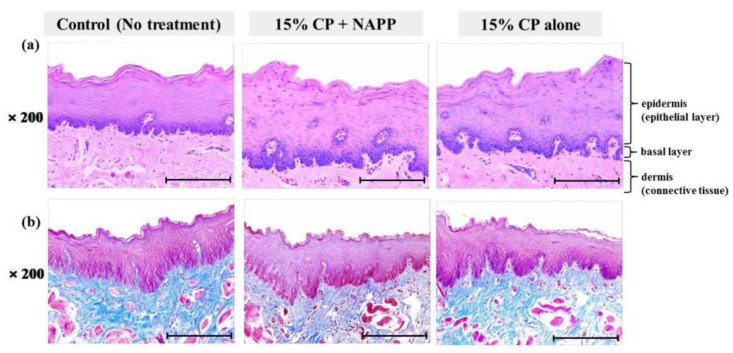
Representative histological image of buccal mucosal tissue of each group with (**a**) H&E and (**b**) MST staining. Magnification ×200.

**Figure 8 ijerph-18-04714-f008:**
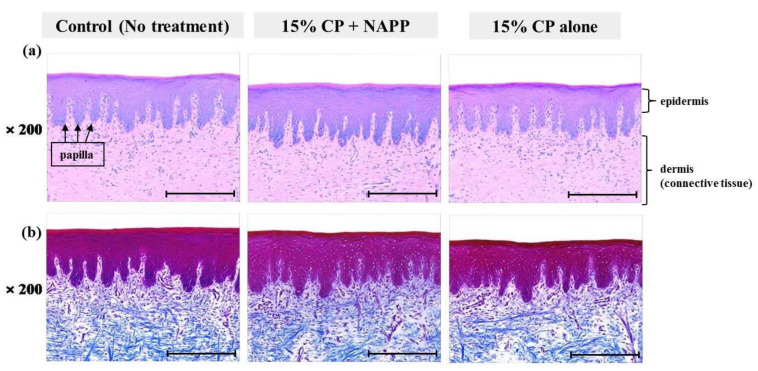
Representative histological image of hard palate tissue of each group with (**a**) H&E and (**b**) MST staining. Magnification ×200.

**Figure 9 ijerph-18-04714-f009:**
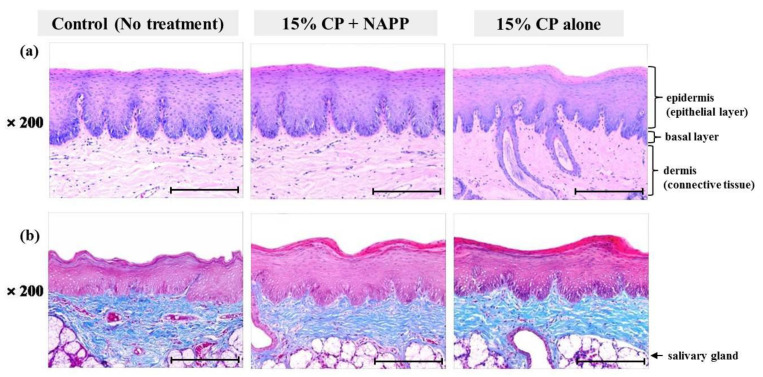
Representative histological image of soft palate tissue of each group with (**a**) H&E and (**b**) MST staining. Magnification ×200.

## Data Availability

The data presented in this study are available on request from the corresponding author. The data are not publicly available due to privacy restrictions.

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
