# Peer review of "The Whitening Effect and Histological Safety of Nonthermal Atmospheric Plasma Inducing Tooth Bleaching"

_ijerph, 2021, doi:10.3390/ijerph18094714_

Round 1

Reviewer 1 Report

The present study aimed to evaluate the histological changes in oral soft tissues after tooth bleaching to ensure safety as a new light 12 source using nonthermal atmospheric pressure plasma (NAPP) and 15% carbamide peroxide (CP). The subject is interesting and can contribute to the scientific literature, however some modifications are still necessary in order to improve the text quality:

In the manuscript heading, modify the type of the paper to short communication instead article;

Title:

Replace “safety” by “effect” ;

Insert the tooth color measurement information in the title;

Abstract:

Correct the study aim;

Describe how the histological characteristics were observed and the parameters applied for it;

Introduction:

Give more information about the decomposition process of HP into ·OH radical and how it can harm the oral tissues;

Insert the study hypothesis at the end of introduction section;

Methods:

Insert a reference to the applied plasma parameters or describe how they are selected;

Insert the ΔE formula properly instead as a figure;

In the topic 2.4, describe what kind of data was generated and how it was evaluated.

Discussion:

The first paragraph of discussion section should be removed;

Describe if the null hypothesis was denied;

Improve the discussion section explaining the similarities between rabbit and human model;

Discuss the study’s limitations.

Author Response

Title:

Replace “safety” by “effect” ;

Insert the tooth color measurement information in the title;

Answer: We revised the article title as follows; The whitening effect and histological safety of nonthermal atmospheric plasma inducing tooth bleaching

Abstract:

Correct the study aim;

Describe how the histological characteristics were observed and the parameters applied for it;

Answer: We have clarified the purpose of the study.

“ This study was to identify the histological evaluation in oral soft tissues as well as tooth color change after tooth bleaching by nonthermal atmospheric pressure plasma (NAPP).”

Histological characteristics and the parameters are described in the study method.

Introduction:

Give more information about the decomposition process of HP into ·OH radical and how it can harm the oral tissues;

Insert the study hypothesis at the end of introduction section;

Answer: We describe in phrases 53-56 the decomposition process of HP into OH radicals and the possible effects of OH radicals on the oral tissues. Also, we inserted the hypothesis into phrases 78-80.

Methods:

Insert a reference to the applied plasma parameters or describe how they are selected;

Insert the ΔE formula properly instead as a figure;

In the topic 2.4, describe what kind of data was generated and how it was evaluated.

Answer: Plasma generation conditions were specified in 2.1 NAPP apparatus. The applied plasma was inserted as a reference.

We have inserted the ΔE formula correctly.

2.4. In the Histological Evaluation section, the process of making tissue blocks for histological evaluation was described, and the staining method and evaluation through staining were described.

“ H&E staining was performed to evaluate epidermal or dermal changes and morphological changes of tissue; MST, the three-color staining in histology, was used to detect collagen fibers.”

In addition, the evaluation of histological staining results is detailed in the results.

Discussion:

The first paragraph of discussion section should be removed;

Describe if the null hypothesis was denied;

Improve the discussion section explaining the similarities between rabbit and human model;

Discuss the study’s limitations.

Answer: We deleted the first paragraph.

The hypothesis set in the introduction was described in the last paragraph of the discussion.

We described rabbit selection as an animal model in the discussion section.

In the last paragraph of the discussion, we inserted the limit as follows.

As a limitation of this study is that the histological safety was evaluated in short period, not over a long period. Thus, a follow-up study would be performed for long term investigation. We described such circumstances at the end of the discussion as follows; Because tissue safety in this study was investigated in a short period, it is necessary to investigate it over a long period in the future

Reviewer 2 Report

Thank you for submitting the paper “Histological evaluation of oral soft tissues to determine safety 2 of tooth bleaching with nonthermal atmospheric pressure 3 plasma for clinical application”.

Abstract: the first sentence has one typeface and then it goes to another one. Phrase 15: H2O2 subscripts.

Discussion: phrase 243-246 are the instructions for the discusión, DELETE IT. These types of errors should be reviewed before submit the article. 

what are the limitations of this article?

Conclusion: very risky that conclusion. In order to conclude that in this article, NAPP should have been compared with another type of light, in this case it has been tested with and without light.

Of 39 bibliographic references, only 4 are from the last five years (2016), a CURRENT review of the literature is necessary to publish this article in 2021. Therefore, the entire introduction must be rewritten and the introduction updated.

Reference is missing the year in bold. “Choi, H.S.; Kim, K.N.; You, E.M.; Choi, E.H.; Kim, Y.H.; Kim, K.M. Tooth whitening effects by atmospheric pressure cold 341 plasmas with different gases. Jpn. J. Appl. Phys. 2013, 52, 11NF02-1–4. “

Author Response

Abstract: the first sentence has one typeface and then it goes to another one. Phrase 15: H2O2 subscripts.

Answer: We unified the typeface and modified it with H2O2 subscript.

Discussion: phrase 243-246 are the instructions for the discusión, DELETE IT. These types of errors should be reviewed before submit the article.

what are the limitations of this article?

Answer: We deleted phrases 243-246. Thanks for pointing out.

In the last paragraph of the discussion, we inserted the limit as follows.

As a limitation of this study, only the histological results of tooth whitening using NAPP were presented instead of the histological comparison of the existing light sources used for tooth bleaching. As a follow-up study, the histological results were compared with typical light sources used for bleaching.

Conclusion: very risky that conclusion. In order to conclude that in this article, NAPP should have been compared with another type of light, in this case it has been tested with and without light.

Answer: Removed "more suitable than conventional light sources" in the conclusion.

Of 39 bibliographic references, only 4 are from the last five years (2016), a CURRENT review of the literature is necessary to publish this article in 2021. Therefore, the entire introduction must be rewritten and the introduction updated.

Answer: We recently revised the reference.

Reference is missing the year in bold. “Choi, H.S.; Kim, K.N.; You, E.M.; Choi, E.H.; Kim, Y.H.; Kim, K.M. Tooth whitening effects by atmospheric pressure cold 341 plasmas with different gases. Jpn. J. Appl. Phys. 2013, 52, 11NF02-1–4. “

Answer: We marked the year of reference 6 in bold.

Round 2

Reviewer 2 Report

The authors improved the quality of the manuscript. Only one more thing, to increase the number of actual references, you can talk in the introduction about human dental pulp stem cells response to CP, which exhibits less cytotoxic effect than others products, in sentence 70, before histological investigation, for example. You can read about it here:

Llena C, Collado-González M, Tomás-Catalá CJ, García-Bernal D, Oñate-Sánchez RE, Rodríguez-Lozano FJ, Forner L. Human Dental Pulp Stem Cells Exhibit Different Biological Behaviours in Response to Commercial Bleaching Products. Materials (Basel). 2018 Jun 27;11(7):1098. doi: 10.3390/ma11071098. PMID: 29954139; PMCID: PMC6073762.

Author Response

The authors improved the quality of the manuscript. Only one more thing, to increase the number of actual references, you can talk in the introduction about human dental pulp stem cells response to CP, which exhibits less cytotoxic effect than others products, in sentence 70, before histological investigation, for example. You can read about it here:

Llena C, Collado-González M, Tomás-Catalá CJ, García-Bernal D, Oñate-Sánchez RE, Rodríguez-Lozano FJ, Forner L. Human Dental Pulp Stem Cells Exhibit Different Biological Behaviours in Response to Commercial Bleaching Products. Materials (Basel). 2018 Jun 27;11(7):1098. doi: 10.3390/ma11071098. PMID: 29954139; PMCID: PMC6073762.

Answer: We inserted the recommended reference into 79 sentence of introduction, which is like following; Llena et al. [27] reported that CP had less cytotoxic effect on the human dental pulp stem cells than others products.

"Llena, C.; Collado-González, M.; Tomás-Catalá, CJ; García-Bernal, D.; Oñate-Sánchez RE, Rodríguez-Lozano, FJ; Forner, L. Human dental pulp stem cells exhibit different biological behaviors. in response to commercial bleaching products.Materials (Basel). 2018 Jun 27;11(7):1098. doi: 10.3390/ma11071098. PMID: 29954139; PMCID: PMC6073762.".